# Spectroscopy and Cyclic Voltammetry Properties of SPEEK/CuO Nanocomposite at Screen-Printed Gold Electrodes

**DOI:** 10.3390/nano12111825

**Published:** 2022-05-26

**Authors:** Omolola E. Fayemi, Onkarabile G. Pooe, Funmilola A. Adesanya, Ikechukwu P. Ejidike

**Affiliations:** 1Department of Chemistry, Faculty of Natural and Agricultural Sciences, North-West University, Mafikeng Campus, Private Bag X2046, Mmabatho 2735, South Africa; pooeog@gmail.com (O.G.P.); fadenikesanya@gmail.com (F.A.A.); 2Material Science Innovation and Modelling (MaSIM) Research Focus Area, Faculty of Natural and Agricultural Sciences, North-West University, Mafikeng Campus, Private Bag X2046, Mmabatho 2735, South Africa; 3Department of Chemical Sciences, Faculty of Science and Science Education, Anchor University, Lagos 100278, Nigeria; iejidike@aul.edu.ng; 4Department of Chemistry, College of Science, Engineering and Technology, University of South Africa, Private Bag X6, Florida 1710, South Africa

**Keywords:** sulfonated polyether ether ketone, electrochemical, CuO NPs, nanocomposite, electrode, surface area, cyclic and square wave voltammetry

## Abstract

A successful electrochemical study of copper oxide nanoparticles (CuO NPs), sulfonated poly (ether ether ketone) polymer (SPEEK), and sulfonated polyether ether ketone-copper oxide (SPEEK/CuO) nanocomposite on bare gold electrodes was conducted. The synthesized CuO NPs and SPEEK/CuO nanocomposite were characterized using X-ray diffraction (XRD), ultraviolet-visible spectroscopy (UV–Vis), Fourier transform infrared spectroscopy (FTIR), scanning electron microscopy (SEM), and electron dispersive spectroscopy (EDS). The XRD showed that the diameter of the CuO NPs synthesized was 20.44 nm. The cyclic voltammetry properties of bare screen-print gold, SPEEK and SPEEK/CuO-modified electrodes were assessed in a 5 mM K_3_[Fe(CN)_6_] solution. The electrochemical performance of the fabricated electrodes investigated revealed that CuO NPs improved the electrochemical properties of SPEEK, and the quantum size effect indicated good adsorption by the SPEEK/CuO nanocomposite compared to the SPEEK polymer and the CuO NPs alone. Moreover, the Tafel values indicated the enhanced electrochemical performance of the other electrodes as compared with the SPEEK/CuO nanocomposite. This, therefore, confirmed the successful incorporation of CuO NPs into the SPEEK polymer, as the increased surface area and the interactions between the polymer matrix and CuO fillers improved the electrochemical performance of the electrode.

## 1. Introduction

In recent years, material scientists have shown great interest in the synthesis of transition metal oxide nanoparticles, mainly due to their corresponding bulk counterparts [1]. Metal oxide nanoparticles have shown great potential in medicine, the environmental sciences, energy, food safety, etc. [2]. Metal oxides play a pivotal role in various scientific disciplines such as chemistry, physics, and material science. Scientists can create a great diversity of oxide compounds from metal elements. These oxides can adopt many structural configurations as they have an electronic construction that can behave as semiconductors, metals, or insulators. Metal oxides are used in the micro-engineering of circuits, sensors, piezoelectric devices, fuel cells, and as plating for surface protection against corrosion and catalysts [3,4].

Copper is famously known as an excellent electrical or heat conductor, and it is cheaper than gold and silver, making its synthesis important. Researchers have invested a great deal of interest in copper oxide nanoparticles (CuO NPs) due to their attractive properties, including low cost, non-toxicity, and simple synthesis methodology [5]. CuO NPs, categorized as a transition metal oxide group, are p-type semiconductors. They are characterized by a monoclinic structure, high stability, and antimicrobial activity [6]. CuO NPs have received much attention compared to other oxides of transition metals. However, there are fewer reports on the preparation and characterization of CuO NPs compared to other transition metal oxides, such as ZnO, TiO_2_, TnO_2_, and FeO nanoparticles [7].

The formation mechanisms of CuO NPs show that, at a low reaction time, the shapes of CuO NPs are irregular [8]. Grigore et al. suggested that the particle morphology, size, and crystallinity of CuO NPs depend on the method of synthesis of the nanoparticles and that these properties are essential to their applications in various fields of study [8]. It has been emphasized that the crucial feature is the size of the synthesized nanoparticle, as it allows tailored modelling of their catalytic, biological, and optical properties [7,8,9].

Polyether ether ketone (PEEK) is an essential engineering plastic, and it possesses a thermoplastic resin structure. The repeating unit arrangements of the oxygen-p-phenylene-oxygen-p-phenylene-carbonyl-p-phenylene are responsible for the molecular backbone of PEEK. PEEK owes its excellent processability and flexibility to the carbonyl and ether bonds and its heat resistance to several benzene rings [10]. It is challenging to integrate PEEK with other polymers because of its insoluble and infusible characteristics; hence, the need to modify it to enhance its solubility. It is for this reason that sulfonated polyether ether ketone (SPEEK) has mechanical strength and conductivity [11]. SPEEK can be used in DMFC [12] due to its high mechanical strength, thermal stability, low cost, easy handling, low methanol crossover, and moderate proton conductivity. The raw material for SPEEK, which is PEEK, is a hydrophobic polymer and is not suitable to fabricate as a membrane. Therefore, it is necessary to functionalize PEEK using concentrated sulfuric acid through the sulfonation process. Other proton exchange membranes that have been successfully proposed in recent years are polymer blends and polymer composite membranes [12]. The preparation of such nanocomposite systems may enhance proton conductivity, reduce methanol permeability, and improve mechanical strength [12].

Nanocomposites are hybrid materials synthesized by mixing polymers with inorganic solids at the nanoscale range. They have a high level of performance, and the combination of different composites produces unique properties. Nanocomposites are used as alternatives to overcome the limitations in the stoichiometry and elemental compositional control of micro-and monolithic composites. The three classes of nanocomposites are ceramic matrix nanocomposites (CMNC), metal matrix nanocomposites (MMNC), and polymer matrix nanocomposites (PMNC). These classes are named according to the matrix or material that is used [12,13,14].

In this study, we synthesized the SPEEK/CuO nanocomposite to increase the electrocatalytic property of the SPEEK polymers. The properties of the CuO NPs are compatible with those of the SPEEK polymer, which produced an enhanced electrocatalytic property of the electrode.

## 2. Materials and Methods

The materials used for this study included deionized water, copper nitrate trihydrate (Cu(NO_3_)_2_·3H_2_O; ≥99.0%) purchased from Glassworld, Johannesburg, South Africa, polyether ether ketone (PEEK) powder with a mean particle size of 80 microns and a weight of 100, sulphuric acid and N-methyl-2-pyrrolidine obtained from Sigma-Aldrich, Steinheim, Germany, and sodium hydroxide (NaOH; ≥99.0%) from Merck KGaA, Darmstadt, Germany.

### 2.1. Preparation of CuO NPs

A known weight of Cu(NO_3_)_2_·3H_2_O (4 g) reacted with 1 M of NaOH was dissolved in deionized water. The mixture was vigorously stirred at 70 °C for 2 h. A dark-bluish precipitate was obtained, which was left overnight to separate. The final product, a black precipitate, was washed with deionized water and absolute ethanol three times. The product was dried for 4 h at 65 °C, then calcinated at 400 °C for two h [15].

### 2.2. Preparation of SPEEK and SPEEK/CuO Nanocomposite

The fabrication of SPEEK was performed following the sulphonation of polyether ether ketone (PEEK), as reported by Dong et al., with few modifications [16]. Approximately 5.0 g of PEEK pellets were dissolved in 100 mL of sulphuric acid under a nitrogen atmosphere at room temperature. The solution was stirred vigorously at 55 °C for 5 h and cooled to room temperature by adding ice-cold deionized (DI) water. The grey-white precipitate was repeatedly washed with DI water until the pH became neutral. The final product was dried at 70 °C in an oven, and the obtained product was sulfonated polyether ether ketone (SPEEK). A total of 20 mg of SPEEK was dissolved with N, N-dimethyl formamide, and 40 mg of the CuO NPs was added and left to sonicate at room temperature for 24 h, and the paste formed was washed and allowed to dry. The obtained product was pulverized and stored for further characterization [16].

### 2.3. Characterization of CuO NPs, SPEEK, and SPEEK/CuO Nanocomposite

The UV−vis spectrophotometer (Agilent Technology, Cary 300 series UV−vis spectrometer, Darmstadt, Germany) operated from 250 to 700 nm. A dual-beam measurement mode was utilized at 1 nm resolution and a 200 nm/min scan rate at 25 °C to investigate the successful formation of CuO NPs and the nanocomposite (SPEEK/CuO). Fourier transform infrared (FTIR) (Agilent Technology, Cary 670 series FTIR spectrometer, Darmstadt, Germany) spectroscopy was used to detect the functional groups present in the CuO NPs, SPEEK, and SPEEK/CuO nanocomposite. XRD (Bruker Company, Karlsruhe, Germany) was used to investigate the crystal structure of the synthesized CuO NPs, SPEEK, and SPEEK/CuO. Scanning electron microscopy (SEM) and energy dispersive X-ray (EDX) (JEOL JSM-6610 LV, Dearborn, Peabody, MA, USA) were utilized to study the surface morphology and the chemical composition of the synthesized CuO NPs, SPEEK, and SPEEK/CuO nanocomposite.

### 2.4. Electrode Modification and Electrochemical Analysis

CuO NPs (20 mg), SPEEK (20 mg), and SPEEK/CuO (20 mg each) were dispersed in DMF and ultrasonicated for 48 h. The formed pastes were deposited on SPAu and dried in air to give SPAu/CuO NP, SPAu-SPEEK, and SPAu-SPEEK/CuO NP. Electrochemical studies were carried out on the screen-printed gold working electrode (4 mm diameter), the carbon counter electrode, and the silver reference electrode. The electrochemical properties were conducted using cyclic voltammetry in a −0.2 to 1.0 V window and a scan rate of 25 mV/s.

## 3. Results and Discussion

### 3.1. UV–Vis Analysis

UV−vis absorption spectroscopy is an interesting technique for the characterization of CuO NPs, SPEEK, and the SPEEK/CuO nanocomposite, and the spectra are shown in Figure 1a–c. The CuO NPs UV absorption peak usually appears in the ultra-violet visible range between 280 to 360 nm [16]. Figure 1a shows the CuO NPs UV absorption peak at approximately 298 nm, which agrees with the literature on CuO NPs UV [2,5,17]. Figure 1b is the UV absorption spectrum for the SPEEK, with an absorption peak at around 300 nm corresponding to the π→π* transitions for the phenyl rings and the n→π* transitions for the C=O and -SO_3_H groups, respectively. [4,14,18]. As observed in Figure 1c, the CuO NPs characteristic peak is at 378 nm in the SPEEK/CuO NPs spectra, confirming the successful formation of the SPEEK/CuO NPs nanocomposites. The absorption band at around 299 nm in the nanocomposite spectra could be due to the π absorption of the functional groups in the polymers [19].

### 3.2. FTIR Analysis

Fourier transformation infrared (FTIR) spectroscopy verified and provided information on various functional groups present in the compounds and materials. FTIR spectra of CuO NPs, SPEEK, and the SPEEK/CuO nanocomposite were recorded at room temperature from 400 to 4000 cm^−1^ as presented in Figure 2a–c. In the CuO NPs spectrum, the absorption band at 489 cm^−1^ Figure 2a is typical of the ν(Cu-O) stretching vibration [14]. Prakash et al. showed that the characteristic peaks of the CuO NPs are in the 400–650 cm^−1^ range on the FTIR spectra, and the additional peaks in the range of 1380–1390 cm^−1^ indicate Cu^2+^–O^2−^ stretching [20]. These bands show the characteristics of the monoclinic CuO NPs. Absorption peaks observed at 3489, 3021, and 2978 cm^−1^ correspond to ν(–OH) stretching of the phenol group and ν(–C-H) symmetric and asymmetric vibrations, which agrees with the literature [4,12,14,17].

The FTIR spectra of SPEEK and the SPEEK/CuO nanocomposite in Figure 2b,c displayed similar bands. The absorption peak at 1639 cm^−1^ in Figure 2b corresponds to the carbonyl group of the SPEEK. The absorption in the 1591 and 1497 cm^−1^ bands correspond to ν(–C=C) stretching of the phenyl rings [12,17]. The intense peaks at 1089 cm^−1^ and 1137 cm^−1^ can be attributed to the symmetric and asymmetric stretching of O=S=O, while the peak at 1371 cm^−1^ can be assigned to the stretching of νs [–SO_3_H], as is reported in the literature [21,22]. Figure 2b shows the FTIR spectra for sulfonated PEEK; three bands were observed and assignable to the sulfonic acid ν(–SO_3_H) groups in the wavelengths 1011, 1102, and 1230 cm^−1^ [17]. The ν(O=S=O) was observed at 1221 cm^−1^, the symmetric stretch was found at 1072 cm^−1^, and the ν(S-O) stretch was observed at 773 cm^−1^ [13]. The absorption peaks at 494 and 626 cm^−1^ in the CuO NPs and SPEEK/CuO nanocomposites correspond to Cu-O stretching. This suggests modification of the SPEEK polymer properties caused by the presence of CuO NPs and confirmed the successful formation of the SPEEK/CuO nanocomposite.

### 3.3. XRD Analysis

The XRD diffraction patterns for CuO NPs, SPEEK, and the SPEEK/CuO NPs nanocomposite were recorded from 20° to 90° as shown in Figure 3a–c. The XRD peaks show the monoclinic phase formation of the synthesized CuO NPs (Figure 3a). The diffraction peaks of the CuO NPs were observed at 2 theta (θ) values of Bragg’s angle for 32.72° (110), 35.73° (111), 38.98° (200), 49.78° (202), and 61.92° (113), with their corresponding Miller indices (h k l) in parentheses, which are similar to those reported in the literature [16,20,23,24,25]. With Scherer’s equation, the average crystalline sizes for the CuO NPs were measured to be 20.44 nm, which is close to the reported values in the range of 14–25 nm.
(1)$$D=\frac{K\lambda }{\beta cos\theta }$$
where *D* is the average crystalline size, *K* is the Scherer’s constant with ranges from 0.68 to 2.08, and 0.94 for a spherical crystalline with cubic symmetry, *λ* is the X-ray wavelength, CuKα = 1.5406 Å, *β* is the full width at the half maximum (FWHM) peak, and *θ* is the Bragg’s diffraction angle [26]. The XRD pattern of the SPEEK exhibited a peak at 19.20° (Figure 3b). The diffraction peak obtained in the XRD pattern (Figure 3c) for the SPEEK/CuO nanocomposite with a similar pattern to CuO NPs with an additional peak at around 19.50° shows the successful incorporation of the CuO NPs in the SPEEK polymer lattice, which led to the positive synthesis of the SPEEK/CuO NPs nanocomposite.

### 3.4. SEM and EDX Analysis

Scanning electron microscopy (SEM) and energy dispersive X-ray (EDX) were used to obtain the microstructural information of the CuO NPs, SPEEK, and SPEEK/CuO nanocomposite. Figure 4 shows the SEM images and the corresponding EDS. Figure 4a portrays the morphology of CuO NPs, which shows agglomerated shards-like particles, which could be attributed to the time and temperature allocated to the calcination process. Figure 4c shows the SEM image of the SPEEK polymer, which exhibits an irregular shape, and Figure 4e shows the morphology of the SPEEK/CuO nanocomposite, which has a flat surface with cracks on it, thus indicating that the CuO NPs merged with the SPEEK polymer. The nanocomposite appeared to be clustered together with traces of an agglomerated shards-like shape [13,18,24,25].

The EDX analysis of the CuO NPs shows high amounts of Cu elements, with high levels of C, signifying the presence of carbonaceous materials (Figure 4b). The atomic ratio of Cu to O is 2:1, which suggests the presence of Cu_2_O [20]. The SPEEK EDX shows high amounts of C (64.74%) (Figure 4d). As regards the elemental composition of the SPEEK/CuO NPs nanocomposites, it was observed that the Cu element replaced some of the C element in the polymer matrix, with the carbon being reduced to (27.88%), thus indicating the successful synthesis of the SPEEK/CuO nanocomposite (Figure 4f).

### 3.5. Electrochemical Studies Using Cyclic Voltammetry

#### 3.5.1. Electrochemical Characterization of the Bare and Fabricated Electrodes

The electrochemical behaviour of the unmodified (SPAuE) and modified electrodes (SPAuE-CuO, SPAuE-SPEEK, and SPAuE-SPEEK-CuO) was investigated by cyclic voltammetry using a 5 mM K_3_Fe[(CN)]_6_ redox probe prepared in a 0.1 M phosphate buffer solution (PBS) of pH 7, at a scan rate of 25 mV/s, within a potential window of −0.2–1.0 V. Figure 5 shows the comparative cyclic voltammogram recorded at the electrode surfaces. Redox peaks corresponding to the reversible redox reaction of K_3_Fe(CN)_6_ were noticed at all the electrodes. SPAuE-SPEEK-CuO exhibited a pair of redox peaks, with the second peak (*E*_pa_ = 0.5 V, *E*_pc_ = 0.43 V) being assigned to the nanocomposite (SPEEK-CuO), indicating the successful modification of the electrode. After the separate modification of SPAuE with the CuO nanoparticles and SPEEK, a decrease in the peak current was observed, suggesting electron transfer inhibition at the electrodes. A further reduction was observed on the nanocomposite-modified electrode, indicating electron transfer resistance enhancement. This could be ascribed to the blocking of the electrode surface area leading to poor electron transport [27]. A summary of the parameters determined from the cyclic voltammetric measurement on the electrodes is presented in Table 1. The anodic-to-cathodic peak potential separation (∆Ep) is greater than 0.0591 V, suggesting a quasi-reversible reaction on the electrodes.

#### 3.5.2. Effects of Varying Scan Rates

To establish the type of electrode reaction and the kinetics of the electrode surfaces in relation to the electrochemical response of the probe solution, the impact of scan rate variation on the current response of the redox probe was assessed in the range of 25–275 mV/s in 5 mM K3[Fe (CN)6] on the bare SPAuE, SPAuE-CuO, SPAuE-SPEEK, and SPAuE-SPEEK/CuO electrodes. Figure 6A–D represents the CVs achieved on the respective electrode surfaces, displaying an increased redox peak potential and current as the scan rate increases. The shifting of the anodic peak potential to the right and the cathodic peak to the left negative was also noticed with an increasing scan rate, suggesting an efficient mass transfer between the electrodes [28].

Figure 7 shows the linear plots of the peak current versus the square root of the scan rate (*v*^1/2^), with a corresponding correlation coefficient (R^2^) of 0.99, indicating a diffusion-controlled electrode electron transfer reaction. The obtained linear relations are as follows:*I*_pa_ = 401.07 *v*^1/2^ − 15.930 (R^2^ = 0.99748); *I*pc = 461.37 *v*^1/2^ +15.930 (R^2^ = −0.99281) SPAuE(2)
*I*_pa_ = 244.41 *v*^1/2^ + 3.0537 (R^2^ = 0.99967); *I*pc = 265.73 *v*^1/2^ −8.7565 (R^2^ = −0.9967) SPAuE-CuO(3)
*I*_pa_ = 32.221 *v*^1/2^ −3.0114 (R^2^ = 0.99623); *I*_pc_ = 20.557 *v*^1/2^ + 3.1221 (R^2^ = −0.99544) SPAuE-SPEEK(4)
*I*_pa_ = 28.580 *v*^1/2^ − 4.4370 (R^2^ = 0.99455); *I*_pc_ = 49.384 *v*^1/2^ + 4.6867 (R^2^ = −0.99566) SPAuE-SPEEK-CuO(5)

The electroactive surface areas on applying Equation (6) were 3.92, 2.39, 0.31, and 0.27 cm^2^ for the SPAuE, SPAuE-CuO, SPAuE-SPEEK, and SPAuE-SPEEK-CuO electrode surfaces, respectively.
(6)$$I_{p}=\left(2.69\times 10^{5}\right)n^{\frac{3}{2}}AD^{\frac{1}{2}}Cv^{\frac{1}{2}}$$
where *I_p_* is the current in amperes, *n* is the number of electrons transferred; *A* is the active surface area (cm^2^), *D* is the diffusion coefficient (cm^2^/s), *C* is the concentration (mol/cm^3^), *v*^1/2^ is scan rate (V/s); for the probe used, *n* is equal to 1.

Equations (7)–(10) show the obtained linear equation for the various electrodes.
*I*_pa_ = 608.99 *v* + 43.975 (R^2^ = 0.97953); *I*pc = −694.76 *v* − 63.436 (R^2^ = −0.96682) SPAuE(7)
*I*_pa_ = 344.12 *v* + 42.023 (R^2^ = 0.99002); *I*pc = −376.44 *v* − 50.781 (R^2^ = −0.99375) SPAuE-CuO(8)
*I*_pa_ = 45.87 *v* + 2.0506 (R^2^ = 0.99752); *I*pc = −29.29 *v* − 2.0506 (R^2^ = −0.99753) SPAuE-SPEEK(9)
*I*_pa_ = 42.29 *v* + 0.0479 (R^2^ = 0.99427); *I*pc = −72.91 *v* − 2.8318 (R^2^ = −0.99327) SPAuE-SPEEK-CuO(10)

The graphs of peak current versus scan rate for the redox peak displayed a linear pattern, as shown in Figure 8. Based on the slope, the electrochemical surface coverage concentration of the redox probe on the electrode surfaces was estimated to be 5.15 × 10^−3^, 2.91 × 10^−3^, 3.88 × 10^−4^ and 3.58 × 10^−4^ mol/cm^3^ for SPAuE, SPAuE-CuO, SPAuE-SPEEK and SPAuE-SPEEK-CuO accordingly, employing Equation (11).
(11)$$\;\;\;\;\;\;\;\;\;\;\;\;\;\;\;\;\;\;\;\;\;\;\;I_{p}=\frac{n^{2}F^{2}\Gamma A}{4RT}$$

Figure 9 presents the graph of peak potentials (*E*_p_) versus the logarithm of the scan rate (*v*). It shows two linear lines with a slope equivalent to $=-\frac{2.3RT}{\alpha nF}\;\mathrm{and}\;\frac{2.3\mathrm{RT}}{\left(1-\alpha \right)nF}$ corresponding, to the cathodic and anodic peak, according to Laviron [29]. The linear Equations (12)–(15) obtained at the different electrodes, together with the correlation coefficients, are given as follows:*E*_pa_ = 0.15495 log v + 0.01897 (R^2^ = 0.99714); *E*_pc_ = −0.12778 log ν +0.19951 (R^2^ = −0.98939) SPAuE(12)
*E*_pa_ = 0.1284 log v + 0.01897 (R^2^ = 0.99714); *E*_pc_ = −0.12778 log ν +0.19951 (R^2^ = −0.98939) SPAuE-CuO(13)
*E*_pa_ = 0.15495 log v + 0.04496 (R^2^ = 0.98124); *E*_pc_ = −0.11999 log ν + 0.22322 (R^2^ = −0.92254) SPAuE-SPEEK(14)
*E*_pa_ = 0.05504 log v + 0.0663 (R^2^ = 0.98119); *E*_pc_ = −0.0492 log ν + 0.16926 (R^2^ = −0.98384) SPAuE-SPEEK-CuO(15)

The estimated charge transfer coefficients (α) for approximately one-electron transfer at the electrode based on the slope obtained (Figure 9A–D) were 1 (SPAuE), 0.51 (SPAuE-CuO), 0.52 (SPAuE-SPEEK), and 0.52 (SPAuE-SPEEK-CuO.

The Tafel values “b” were computed as 0.309, 0.256, 0.188, and 0.110 V/dec for SPAuE, SPAuE-CuO, SPAuE-SPEEK, and SPAuE-SPEEK-CuO, respectively, using Equation (16). The Tafel values were greater than the theoretical values (0.118 v/dec at all the electrodes except for SPAuE-SPEEK-CuO, which had a value of less than 0.118 v/dec, suggesting an adsorption reaction) [30].
(16)$$Ep=\left(\frac{b}{2}\right)\mathrm{logv}\;+\;\mathrm{constant}$$

## 4. Conclusions

Upon the successful synthesis of the CuO NPs, the SPEEK/CuO nanocomposite, the samples were subjected to various characterization techniques, and their electrochemical properties were investigated using the cyclic voltammetry technique. The Tafel value confirmed that the presence of the CuO NPs had a positive effect on the SPEEK polymer as the Tafel value on the SPEEK/CuO nanocomposite was approximately three times lower than that of the polymer alone. Furthermore, from this Tafel value, we can conclude that there was enhanced adsorption on our nanocomposite compared to the polymer and the CuO NPs alone, even though the bare gold electrode had a higher current response than all the fabricated electrodes. This, therefore, confirmed our predictions regarding the results of this study.

## Figures and Tables

**Figure 1 nanomaterials-12-01825-f001:**
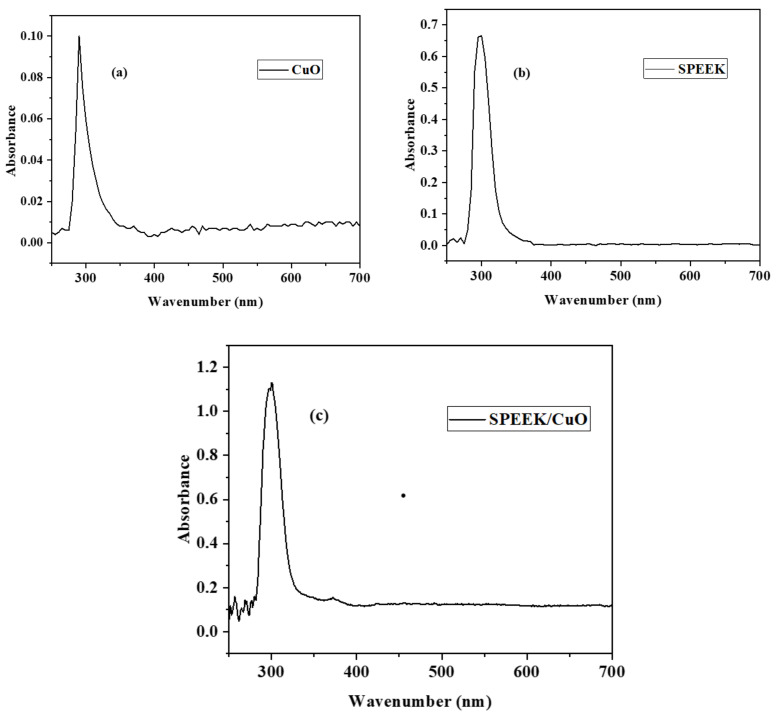
UV–vis spectra of (**a**) CuO NPs, (**b**) SPEEK polymer, and (**c**) SPEEK/CuO NPs nanocomposite.

**Figure 2 nanomaterials-12-01825-f002:**
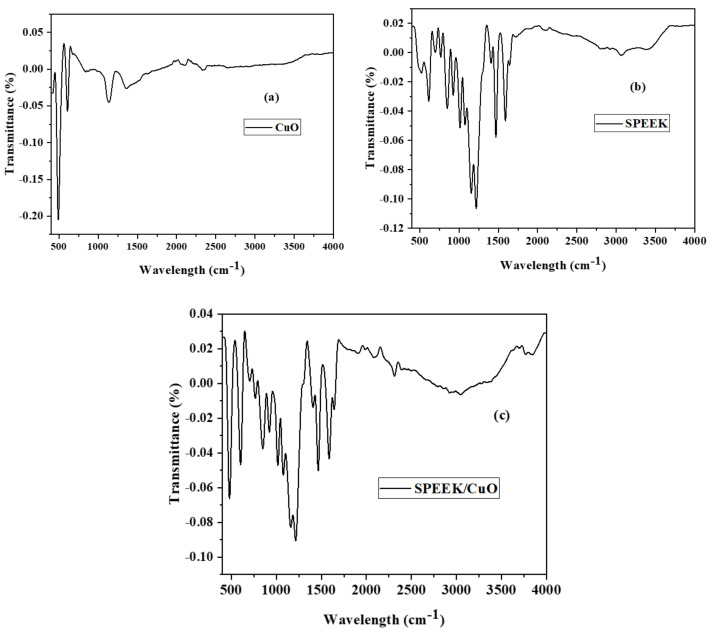
FTIR spectra of CuO NPs (**a**), SPEEK (**b**), and SPEEK/CuO NPs (**c**).

**Figure 3 nanomaterials-12-01825-f003:**
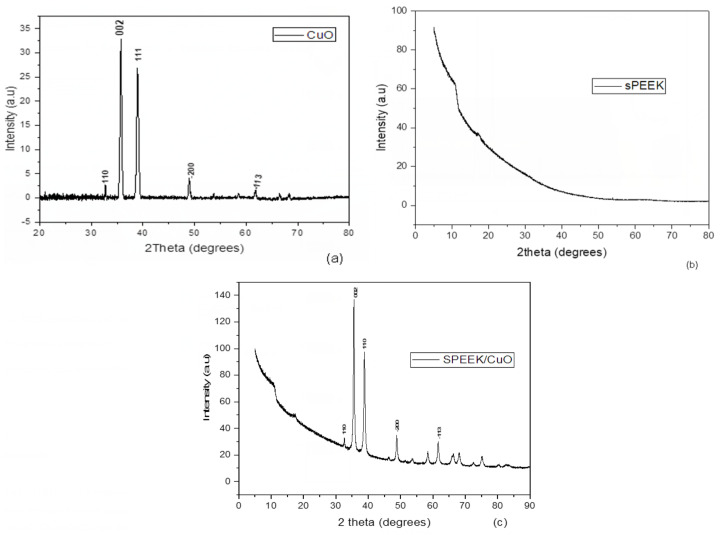
XRD spectra of (**a**) CuO NPs, (**b**) SPEEK, and (**c**) SPEEK/CuO NPs.

**Figure 4 nanomaterials-12-01825-f004:**
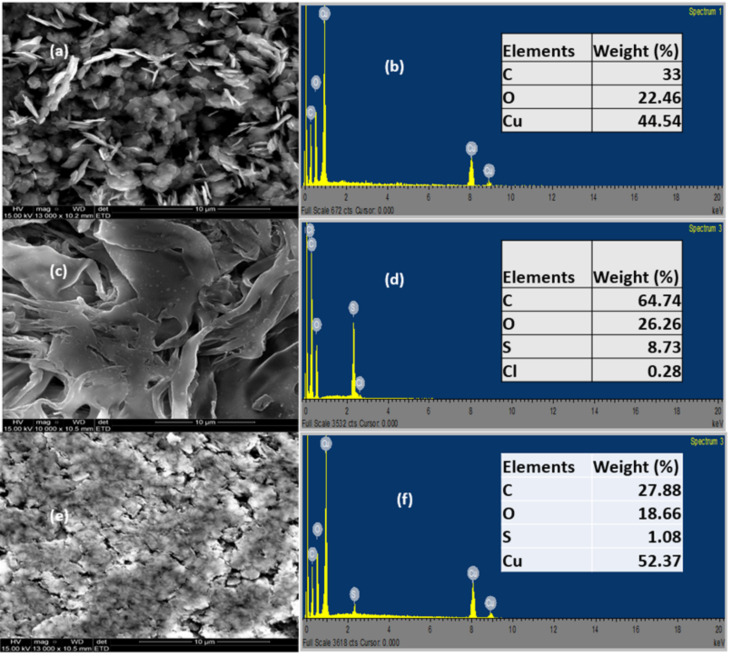
SEM images (**a**,**c**,**e**) and EDX analysis (**b**,**d**,**f**) for the CuO NPs, SPEEK, and SPEEK/CuO nanocomposite, respectively.

**Figure 5 nanomaterials-12-01825-f005:**
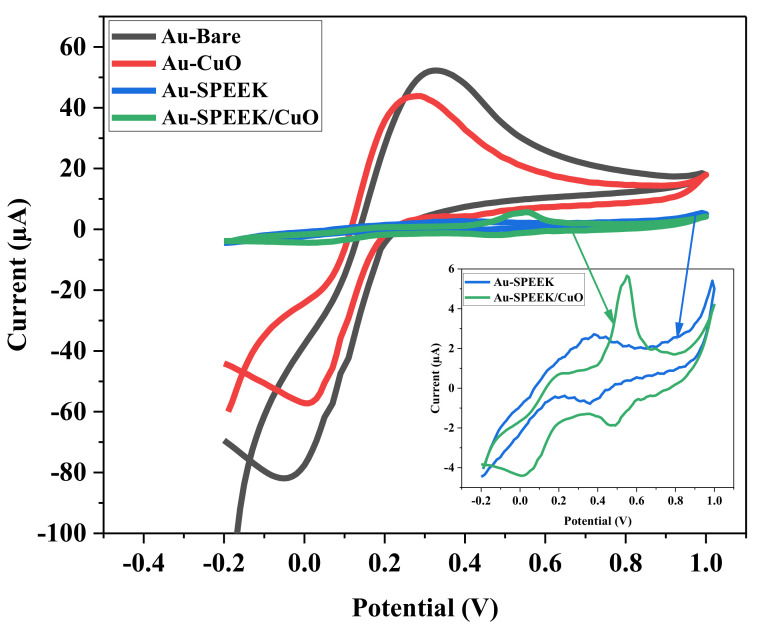
Comparative cyclic voltammogram of the fabricated electrodes (Au-Bare, Au-CuO NPs, Au-SPEEK, and Au-SPEEK/CuO NPs) in 5 mM K_3_[Fe(CN)_6_] prepared in 0.1 M PBS (pH 7.4) at a scan rate of 25 mV/s.

**Figure 6 nanomaterials-12-01825-f006:**
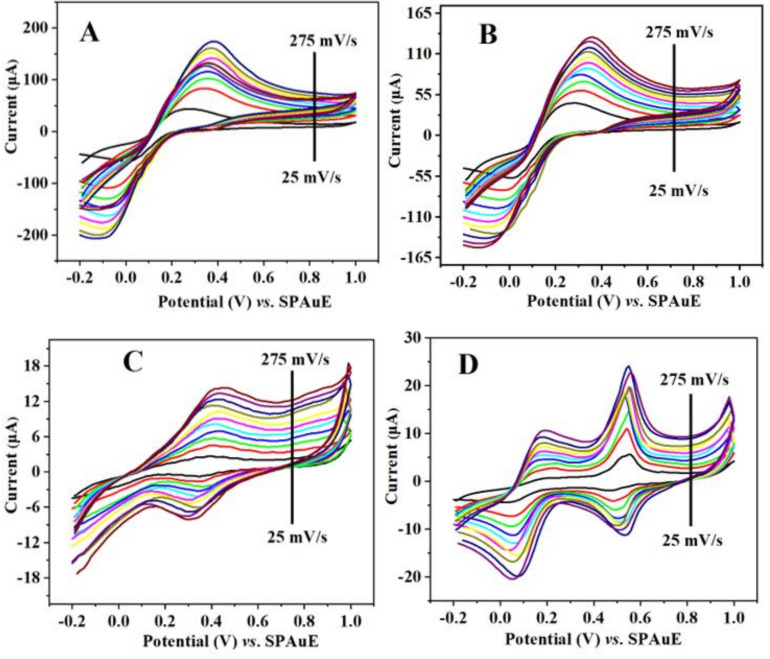
Cyclic voltammogram at scan rate ranging from 25 to 275 mV/s of 5 mM K_3_[Fe(CN)_6_] in 0.1 M PBS for (**A**) SPAuE (**B**) SPAuE-CuO (**C**) SPAuE-SPEEK, and (**D**) SPAuE-SPEEK-CuO.

**Figure 7 nanomaterials-12-01825-f007:**
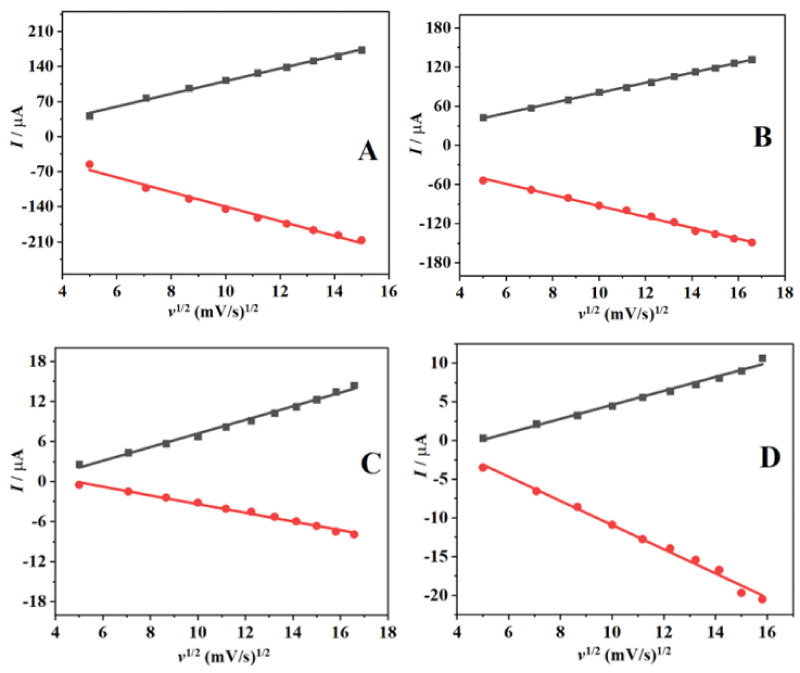
The linear relationship between *I*_p_ and *v*^1/2^ for (**A**) SPAuE, (**B**) SPAuE-CuO, (**C**) SPAuE-SPEEK, and (**D**) SPAuE-SPEEK-CuO.

**Figure 8 nanomaterials-12-01825-f008:**
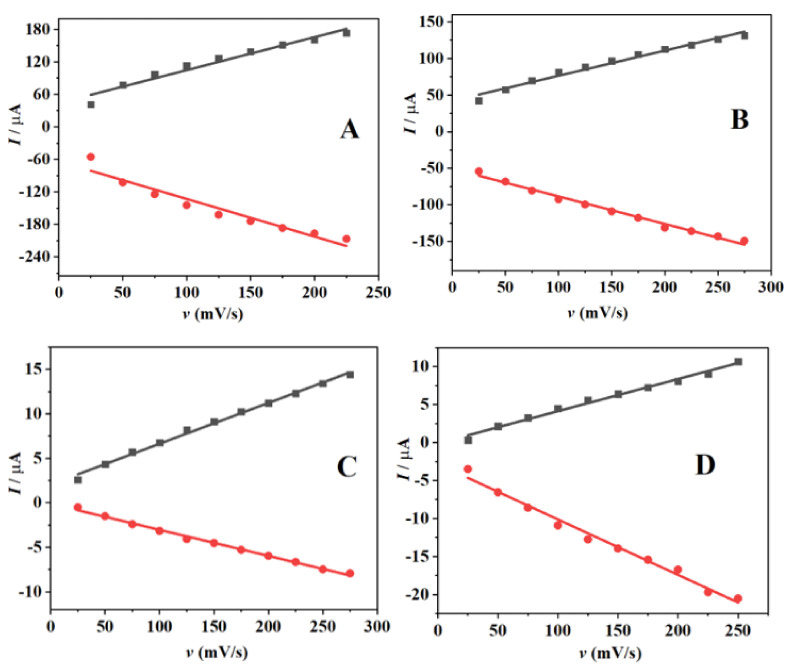
A linear relationship between *I*_p_ and *v* for (**A**) SPAuE, (**B**) SPAuE-CuO, (**C**) SPAuE-SPEEK, and (**D**) SPAuE-SPEEK-CuO.

**Figure 9 nanomaterials-12-01825-f009:**
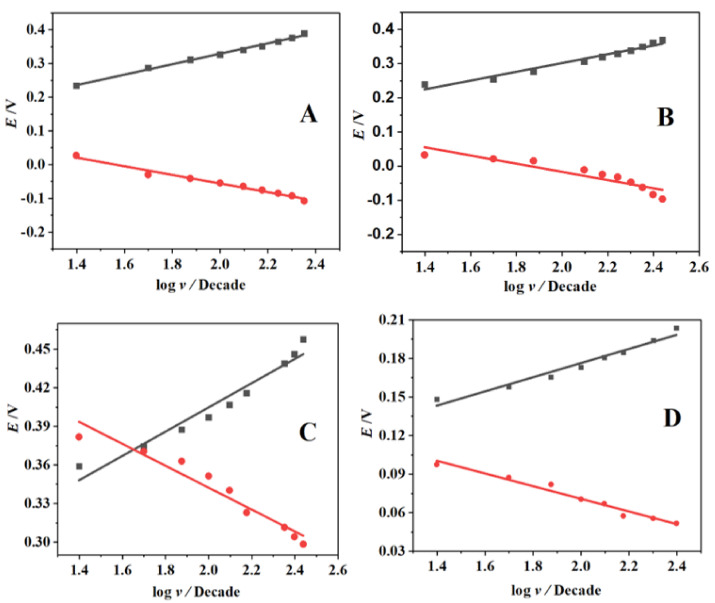
The relationship between E_p_ and the logarithm of the scan rate (*v)* for (**A**) SPAuE, (**B**) SPAuE-CuO, (**C**) SPAuE-SPEEK, and (**D**) SPAuE-SPEEK-CuO.

**Table 1 nanomaterials-12-01825-t001:** Summary of cyclic voltammetry data of the different electrodes (Au-Bare, Au-CuO NPs, Au-SPEEK, and Au-SPEEK/CuO NPs) at a 25 mV/s scan rate.

Electrodes	Ipa (µA)	Ipc (µA)	Ipa/Ipc	Epa (V)	Epc (V)	ΔEp (V)	Eo (V)
**SPAuE**	52.39	−82.15	0.64	0.33	−0.05	0.38	0.14
**SPAuE-CuO**	44.02	−57.33	0.77	0.28	0.0067	0.27	0.14
**SPAuE-SPEEK**	2.77	−0.84	3.30	0.38	0.36	0.02	0.37
**SPAuE-SPEEK/CuO**	5.73	−1.92	2.98	0.55	0.47	0.08	0.51

## Data Availability

The data presented in this study are available on request from the corresponding author.

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
