# Peer review of "Spectroscopy and Cyclic Voltammetry Properties of SPEEK/CuO Nanocomposite at Screen-Printed Gold Electrodes"

_nanomaterials, 2022, doi:10.3390/nano12111825_

Round 1

Reviewer 1 Report

This manuscript reported the investigation of the SPEEK/CuO nanocomposite at bare gold electrodes. The electrochemical performance of the fabricated electrodes investigated revealed that the surface area of the CuO NPs was higher than the rest of the electrodes used. In addition, a better electrochemical performance was observed for the fabricated composite electrodes.

I consider the content of this review will meet the reading interests of the readers of the Nanomaterials journal. However, the authors still need to make certain modifications based on my following comments. Therefore, I suggest giving a minor revision and the authors need to address the issues raised in these comments. This could be meaningful work after revision.

  1. Please pay attention to grammar and spelling problems, especially the missing or redundant definite articles. I suggest double-checking. I will point out several examples, but unfortunately, I cannot point out all of them.

For example:

â‘  Line 49, ‘Researchers have invested ... due to their attractive properties, including low cost, non-toxicity and simple synthesis methodology, making them more desirable [5]’;

â‘¡ Line 129, the ‘3’ after ‘-0.2 to 1.0 V’ should be deleted.

â‘¢ Line 130, it should be ‘3. Results and Discussion

â‘£ Line 134, ‘The CuO NPs UV absorption peak usually appears in the range of ... which is in agreement with the literature on CuO NPs UV [2,5,19]’ and so on.

  1. The title is a bit confusing. Indeed, the obtained ‘SPEEK/CuO nanocomposite’ is used as the electrode. I suggest at least the word electrode should appear in the title. It should be noted that the SPEEK membrane can be used for fuel cells and flow batteries as the ion-exchange membrane, and sometimes inorganic fillers are also introduced into the polymer matrix for hybrid membranes. That is why the topic may confuse the readers since cyclic voltammetry properties of an ion-exchange membrane seem impossible and makes no sense. But as an electrode, it is meaningful enough.

  1. Line 30, ‘This signifies the successful incorporation of CuO NPs into the SPEEK polymer leading to the synthesis of the SPEEK/CuO nanocomposite.This sentence does not seem to have any meaning, because the first half and the second half say the same meaning. If CuO is incorporated into the polymer successfully, of course, we should obtain a nanocomposite membrane between the polymer and CuO. The Tafel results that indicate better electrochemical performance should be strengthened here, for example, ‘This signifies the successful incorporation of CuO NPs into the SPEEK polymer leading to the improved electrochemical performance due to the special interactions between the polymer matrix and CuO fillers and the increased surface area’.

  1. For the Keywords, ‘nanocomposite’, ‘electrode’, and ‘surface area’ should be added in order to attract a broader readership.

  1. Line 70, ‘PEEK needs to be fabricated to improve its interfacial binding force between two phases by changing the polarity of the molecular segments [11].It is not very clear what the two phases are in PEEK since PEEK just has a hydrophobic phase. Moreover, the potential window range is given for the CV test, but how about the scanning rate?

I consider it should be ‘PEEK needs to be modified to be easily soluble, such as after sulfonation to obtain SPEEK’. In this way, the SPEEK have both the hydrophobic phase (pristine PEEK) and the hydrophilic phase (-SO3H), thus the interfacial binding force between the hydrophobic and hydrophilic phases is improved by modifying the polarity of the molecular segments (Ionics 25.9 (2019): 4219-4229). This issue should be explained better since even by checking the provided reference [11], I cannot understand what the original sentence means exactly.

For the next sentence, why does the composite material become ion-exchangeable? This is due to the phase separation between hydrophobic and hydrophilic phases when the membrane is hydrated. It is not to say every polymer combines/incorporates with -SO3H will make ion-exchange material, but only hydrophobic polymer can realize this target. This issue should also be clarified, or even deleted if the authors consider it is not very related to the electrode content.

  1. Line 85, ‘In this study, we synthesized the SPEEK/CuO nanocomposite to increase the properties of the SPEEK polymers. The properties of the CuO NPs are biocompatible with those of the SPEEK polymer resulting in an enhanced property of the nanocomposite.

Since this is in the final part of the Introduction, for the first sentence, the authors are aiming to increase which properties by making nanocomposites, should be described clearly and directly. This part should be like a summary of the work.

And for the second sentence, is the biocompatible related to this work for the application of electrodes and electrochemical performance? I consider it is enough to mention only 'compatible/compatibility', which means that SPEEK and CuO NPs are not exclusive and can be effectively and evenly combined. And which property is enhanced, should also be clarified.

  1. Line 91, where is the ‘Sulfonated polyether ether ketone (SPEEK)’ purchased? It is not very clear to the readers. How about the degree of sulfonation for the SPEEK? How about the PEEK molecular weight before sulfonation or how about the molecular weight of the SPEEK? This data should be provided, or it is difficult for the readers to repeat the experiments if needed.

  1. Line 104, ‘20 mg of SPEEK was dissolved with N, N-dimethyl formamide, and 40 mg of the CuO NPs were added’. Why is this special mass ratio used throughout the whole manuscript? Is this an optimal value between the two? If not, why not study the effect of filler loading on the electrochemical properties of the composites?

  1. In section 4. Electrode Modification and Electrochemical Analysis, if the ‘screen-printed electrode is used as the working electrode’, what are the counter electrode and reference electrode in the Cyclic Voltammetry test? There needs to be a corresponding detailed explanation here.

  1. Line 165, ‘The intense peaks at 1089, 1137 and 1371 cm-1 are attributed to the C−O−H bending vibrations, phenol or alcohol group [14,19].In the SPEEK structure, there should be no alcohol group. Indeed, 1371 cm-1 should be stretching ofνs [-SO3H]. See the assignments of literature (Electrochimica Acta 309 (2019): 311-325; Physical Chemistry Chemical Physics 10.11 (2008): 1577-1583).

  1. Line 216, ‘it was observed that the Cu element replaced the Cl element on the polymer matrix and thus indicating successful synthesis of the SPEEK/CuO nanocomposite’. Here, it is better to explain where the Cl element is from. Since in SPEEK and CuO, there should be no Cl element. And how does Cu as a metal element replace a halogen element Cl?

  1. For the CV part, the calculation is too much detailed. I suggest simplifying the calculation process appropriately because too much step-by-step calculation makes the whole article look very much like an experimental report.

Author Response

The Editor

Nanomaterials

REVISION OF MANUSCRIPT SUBMITTED FOR PUBLICATION

Manuscript Title: Spectroscopy and Cyclic voltammetry properties of SPEEK/CuO nanocomposite at Screen-Printed Gold Electrodes.

Manuscript ID:  nanomaterials-1714077

We appreciate the reports on our manuscript. As a result of this, we submit a response to the comments on the manuscript for further consideration.

The reviewers' efforts at improving this manuscript are well appreciated. We have carefully considered the comments. The responses to all reviewers' comments are highlighted in yellow in the manuscript. 

Reviewer 1

This manuscript reported the investigation of the SPEEK/CuO nanocomposite at bare gold electrodes. The electrochemical performance of the fabricated electrodes investigated revealed that the surface area of the CuO NPs was higher than the rest of the electrodes used. In addition, a better electrochemical performance was observed for the fabricated composite electrodes.

I consider the content of this review will meet the reading interests of the readers of the Nanomaterials journal. However, the authors still need to make certain modifications based on my following comments. Therefore, I suggest giving a minor revision and the authors need to address the issues raised in these comments. This could be meaningful work after revision. 

Comment 1: Please pay attention to grammar and spelling problems, especially the missing or redundant definite articles. I suggest double-checking. I will point out several examples, but unfortunately, I cannot point out all of them.

Response: Thanks for the comment, grammar and spelling problems including missing and redundant definite articles of the manuscript have been checked and addressed accordingly.

For example:

â‘  Line 49, ‘Researchers have invested ... due to their attractive properties, including low cost, non-toxicity and simple synthesis methodology, making them more desirable [5]’;

Response: Thanks for your observation. The sentence has been rephrased see yellow highlight on page 2 lines 51-53

â‘¡ Line 129, the ‘3’ after ‘-0.2 to 1.0 V’ should be deleted.

Response: Thanks the “3” after -0.2 to 1.0 V has been removed. See yellow highlight on page 3 line 134

â‘¢ Line 130, it should be ‘3. Results and Discussion

Response: The title of the session has been changed to “Results and Discussion. See yellow highlight page 3, line 136

â‘£ Line 134, ‘The CuO NPs UV absorption peak usually appears in the range of ... which is in agreement with the literature on CuO NPs UV [2,5,19]’ and so on.

Response: Thank you for the observation. The grammatical and spelling errors have been corrected accordingly. The concerns raised regarding lines 49, 129, 130 and 134 have been addressed (kindly find corrections in yellow highlights).

Comment 2: The title is a bit confusing. Indeed, the obtained ‘SPEEK/CuO nanocomposite’ is used as the electrode. I suggest at least the word ‘electrode’ should appear in the title. It should be noted that the SPEEK membrane can be used for fuel cells and flow batteries as the ion-exchange membrane, and sometimes inorganic fillers are also introduced into the polymer matrix for hybrid membranes. That is why the topic may confuse the readers since cyclic voltammetry properties of an ion-exchange membrane seem impossible and makes no sense. But as an electrode, it is meaningful enough.

Response: The electrode was modified with a nanocomposite of the SPEEK/CuO and has been included in the title as suggested (see the yellow highlight on page 1).

Comment 3: Line 30, ‘This signifies the successful incorporation of CuO NPs into the SPEEK polymer leading to the synthesis of the SPEEK/CuO nanocomposite.This sentence does not seem to have any meaning, because the first half and the second half say the same meaning. If CuO is incorporated into the polymer successfully, of course, we should obtain a nanocomposite membrane between the polymer and CuO. The Tafel results that indicate better electrochemical performance should be strengthened here, for example, ‘This signifies the successful incorporation of CuO NPs into the SPEEK polymer leading to the improved electrochemical performance due to the special interactions between the polymer matrix and CuO fillers and the increased surface area’.

Response: Line 30 - Thank you for the observation. The sentence has been reconstructed using your refined input. Thank you. (See the yellow highlight).

Comment 4: For the Keywords, ‘nanocomposite’, ‘electrode’, and ‘surface area’ should be added in order to attract a broader readership.

Response: The keywords have been modified as suggested ((kindly find corrections in yellow highlights).

Comment 5: Line 70, ‘PEEK needs to be fabricated to improve its interfacial binding force between two phases by changing the polarity of the molecular segments [11].It is not very clear what the two phases are in PEEK since PEEK just has a hydrophobic phase. Moreover, the potential window range is given for the CV test, but how about the scanning rate?

I consider it should be ‘PEEK needs to be modified to be easily soluble, such as after sulfonation to obtain SPEEK’. In this way, the SPEEK have both the hydrophobic phase (pristine PEEK) and the hydrophilic phase (-SO3H), thus the interfacial binding force between the hydrophobic and hydrophilic phases is improved by modifying the polarity of the molecular segments (Ionics 25.9 (2019): 4219-4229). This issue should be explained better since even by checking the provided reference [11], I cannot understand what the original sentence means exactly.

For the next sentence, why does the composite material become ion-exchangeable? This is due to the phase separation between hydrophobic and hydrophilic phases when the membrane is hydrated. It is not to say every polymer combines/incorporates with -SO3H will make ion-exchange material, but only hydrophobic polymer can realize this target. This issue should also be clarified, or even deleted if the authors consider it is not very related to the electrode content.

Response: Line 70 – The sentence has been explained clearly. Thanks a lot for the reference. (Kindly find corrections in yellow highlights). The scan rate has also been included.

The next sentence about the composite material becoming ion-exchangeable has been deleted.

Comment 6: Line 85, ‘In this study, we synthesized the SPEEK/CuO nanocomposite to increase the properties of the SPEEK polymers. The properties of the CuO NPs are biocompatible with those of the SPEEK polymer resulting in an enhanced property of the nanocomposite.

Since this is in the final part of the Introduction, for the first sentence, the authors are aiming to increase which properties by making nanocomposites, should be described clearly and directly. This part should be like a summary of the work.

And for the second sentence, is the ‘biocompatible’ related to this work for the application of electrodes and electrochemical performance? I consider it is enough to mention only 'compatible/compatibility', which means that SPEEK and CuO NPs are not exclusive and can be effectively and evenly combined. And which property is enhanced, should also be clarified.

Response: Line 85 – The property we aimed at increasing has been spelt out in the first sentence on line 85 also the biocompatibility in the second sentence has been replaced with compatible and the enhanced property clarified. (Kindly find corrections in yellow highlights).

  1. Line 91, where is the ‘Sulfonated polyether ether ketone (SPEEK)’ purchased? It is not very clear to the readers. How about the degree of sulfonation for the SPEEK? How about the PEEK molecular weight before sulfonation or how about the molecular weight of the SPEEK? This data should be provided, or it is difficult for the readers to repeat the experiments if needed.

Response:  Thanks for your brilliant observation and comment. The supplier of the PEEK used for the study has been included in the section 2 line 98, page 3, and also section 2.2 line 113, page 3. (kindly see the yellow highlight on this page and sections identified)

  1. Line 104, ‘20 mg of SPEEK was dissolved with N, N-dimethyl formamide, and 40 mg of the CuO NPs were added’. Why is this special mass ratio used throughout the whole manuscript? Is this an optimal value between the two? If not, why not study the effect of filler loading on the electrochemical properties of the composites?

 Response: Thanks for your comment. The mass used was the optimized mass with better electrochemical response.

Comment 9: In section 4. Electrode Modification and Electrochemical Analysis, if the ‘screen-printed electrode is used as the working electrode’, what are the counter electrode and reference electrode in the Cyclic Voltammetry test? There needs to be a corresponding detailed explanation here.

Response: The counter and reference electrode has been included. (Kindly find correction in yellow highlights).

Comment 11: Line 165, ‘The intense peaks at 1089, 1137 and 1371 cm-1 are attributed to the C−O−H bending vibrations, phenol or alcohol group [14,19].In the SPEEK structure, there should be no alcohol group. Indeed, 1371 cm-1 should be stretching of νs [-SO3H]. See the assignments of literature (Electrochimica Acta 309 (2019): 311-325; Physical Chemistry Chemical Physics 10.11 (2008): 1577-1583).

 Response: The peak assignment has been corrected to align with the literature. Thank you for the references. (Kindly find correction in yellow highlights).

Comment 11: Line 216, ‘it was observed that the Cu element replaced the Cl element on the polymer matrix and thus indicating successful synthesis of the SPEEK/CuO nanocomposite’. Here, it is better to explain where the Cl element is from. Since in SPEEK and CuO, there should be no Cl element. And how does Cu as a metal element replace a halogen element Cl?

Response: We admit that the Cl was a mistake, it was meant to be C which has been corrected. (Kindly find correction in yellow highlights).

Comment 12: For the CV part, the calculation is too much detailed. I suggest simplifying the calculation process appropriately because too much step-by-step calculation makes the whole article look very much like an experimental report

Response: Corrections have been made as recommended. Thank you.

Reviewer 2 Report

The authors report the synthesis of SPEEK/CuO nanocomposite to improve the properties of the SPEEK polymers. 

The manuscript is within the scope of Nanomaterials. However, before publication some major concerns must be addressed, namely:

  1. The introduction does not reflect the actual state of the art.
  2. The. authors should avoid the use of words such as better since it is not very accurate.
  3. Figures 9B and 9C (both red) do not show a linear relationship.
  4. The main advances reported must be compared and discussed with the information available in the literature.
  5. The english needs to be improved and "better" shoul be replaced by improved or enhanced.

Author Response

The Editor

Nanomaterials

REVISION OF MANUSCRIPT SUBMITTED FOR PUBLICATION

Manuscript Title: Spectroscopy and Cyclic voltammetry properties of SPEEK/CuO nanocomposite at Screen-Printed Gold Electrodes.

Manuscript ID:  nanomaterials-1714077

We appreciate the reports on our manuscript. As a result of this, we submit a response to the comments on the manuscript for further consideration.

The reviewers' efforts at improving this manuscript are well appreciated. We have carefully considered the comments. The responses to all reviewers' comments are highlighted in yellow in the manuscript. 

Reviewer 2

The authors report the synthesis of SPEEK/CuO nanocomposite to improve the properties of the SPEEK polymers. 

The manuscript is within the scope of Nanomaterials. However, before publication some major concerns must be addressed, namely:

1. The introduction does not reflect the actual state of the art.

Response: Thanks for the comment the introduction section has been improved

2. The. authors should avoid the use of words such as better since it is not very accurate.

Response: Thanks for the comment the use of the word ‘better’ has been minimized.

3. Figures 9B and 9C (both red) do not show a linear relationship.

Response: Thanks for the brilliant observation. The figures’ data has been reviewed and represented. See Figures 9B and 9C on page 12 of the manuscript

4. The main advances reported must be compared and discussed with the information available in the literature.

Response: Thanks for the comment the suggestion has been addressed

5. The english needs to be improved and "better" shoul be replaced by improved or enhanced.

Response: Thanks for the comment. The word ‘better’ has been replaced with ‘enhanced’

Round 2

Reviewer 2 Report

All the issues raised were well addressed. Therefore, I support publication in the present form.